# Telomere Maintenance and the cGAS-STING Pathway in Cancer

**DOI:** 10.3390/cells11121958

**Published:** 2022-06-17

**Authors:** Hiroshi Ebata, Tze Mun Loo, Akiko Takahashi

**Affiliations:** 1Department of Biological Sciences, Faculty of Science, The University of Tokyo, Tokyo 113-0033, Japan; hiroshi.ebata@bs.s.u-tokyo.ac.jp; 2Project for Cellular Senescence, Cancer Institute, Japanese Foundation for Cancer Research, Tokyo 135-8550, Japan; tzemunloo@jfcr.or.jp

**Keywords:** DNA damage, telomere, cGAS-STING, cancer, ALT, cellular senescence

## Abstract

Cancer cells exhibit the unique characteristics of high proliferation and aberrant DNA damage response, which prevents cancer therapy from effectively eliminating them. The machinery required for telomere maintenance, such as telomerase and the alternative lengthening of telomeres (ALT), enables cancer cells to proliferate indefinitely. In addition, the molecules in this system are involved in noncanonical pro-tumorigenic functions. Of these, the function of the cyclic GMP–AMP synthase (cGAS)-stimulator of interferon genes (STING) pathway, which contains telomere-related molecules, is a well-known contributor to the tumor microenvironment (TME). This review summarizes the current knowledge of the role of telomerase and ALT in cancer regulation, with emphasis on their noncanonical roles beyond telomere maintenance. The components of the cGAS-STING pathway are summarized with respect to intercell communication in the TME. Elucidating the underlying functional connection between telomere-related molecules and TME regulation is important for the development of cancer therapeutics that target cancer-specific pathways in different contexts. Finally, strategies for designing new cancer therapies that target cancer cells and the TME are discussed.

## 1. Introduction

Cells are constantly exposed to extracellular or intracellular stress, which causes DNA damage. Hayflick and Moorhead [1] observed that primary human cells proliferate a limited number of times. This indicates that the accumulation of DNA damage leads to genome instability and either programmed cell death or cellular senescence. Even without stress, normal eukaryotic cells eventually experience genome instability, because every cell division shortens the telomeres, which protect the chromosomal ends from being recognized as DNA double-strand breaks (DSBs). In addition to genomic DNA, the accumulation of damage in the mitochondrial DNA may result in proliferative defects [2]. In contrast, cancer cells maintain long telomeres to proliferate indefinitely, even though they exhibit the representative hallmarks of genome instability [3,4,5] and high levels of mitochondrial DNA damage caused by reactive oxygen species (ROS) [6,7]. This suggests that the machinery enables telomere elongation and proliferation under conditions of genome instability in the nucleus and mitochondria. Cancer cells overcome the cell proliferation limit by expressing telomerase, which directly elongates telomeres by reverse transcription [8], or the alternative lengthening of telomeres (ALT), which increases the telomere length through homologous recombination (HR) [9]. By combining each telomere maintenance mechanism with an aberrant DNA damage response (DDR), cancer cells have no limitation on cell division and can accumulate irreparable genome mutations by replication stress. In addition, both telomere maintenance mechanisms, telomerase and ALT, have noncanonical roles in mitochondria [10,11]. Therefore, cancer cells are likely to manage nuclear and mitochondrial DNA damage by tolerating DNA mutations, but not allowing DNA recognition systems to pause proliferation.

The characteristic of the rapid accumulation of DNA mutations in cancer cells is a challenge for eliminating cancer cells with chemotherapy. Recently, cancer cells treated with radiation therapy were reported to cause self-DNA damage by the expression of caspase-activated DNase, in order to evade premature mitosis-mediated cell death by triggering the G2 cell-cycle checkpoint [12]. Likewise, genome instability in cancer cells chronically activates the cytosolic DNA sensor mechanism known as the cyclic GMP–AMP synthase (cGAS)-stimulator of interferon genes (cGAS-STING) pathway [13]. Chronic activation of the cGAS-STING pathway leads to a pro-tumorigenic environment of chronic inflammation [14]. Cancer therapy often damages the surrounding cells, such as cancer-associated fibroblasts (CAFs), and turns them into an abnormal state [15]. The abnormal surrounding cells show characteristics similar to cancer cells, such as genome instability and DDR defects, which activate the cGAS-STING pathway. Combining the inflammatory microenvironment induced by therapy and the high accumulation rate of DNA mutations in cancer cells enables a small subset of targeted cancer cells to evade therapy and recur with high proliferation capacity. In this review, we focus on telomere-related DDR between the nucleus and mitochondria in cancer cells. We also discuss cGAS-STING activation in cancer cells and the surrounding normal cells that eventually contribute to cancer progression (Figure 1).

## 2. Telomere and the cGAS-STING Pathway

In eukaryotic cells, exposed chromosomal ends are recognized as DNA DSBs and trigger DDR, which results in genome instability [16,17]. Each chromosomal end is capped with protective DNA regions known as telomeres to prevent misrecognition. Telomeres are repetitive DNA sequences (TTAGGGs in humans) and as the repeats are guanine-rich, they form a secondary structure known as a G-quadruplex, which exerts a protective role for the chromosomal ends [18]. The G-quadruplex recruits a protein complex known as shelterin [19], which increases telomere protection by inducing another conformation of telomeres, the T-loop. The T-loop is a large structure containing an invasion of the 3′-overhang of telomeric DNA into double-strand telomeric DNA [20]. Despite these protective mechanisms for the genome at the telomeres, telomeres shorten following each cell division because DNA polymerase exhibits 5′–3′ activity during DNA replication and cannot replicate the 5′-regions of the telomeres [21]. This telomere shortening induces senescence, which is a state of irreversible cell-cycle arrest and leads to the proliferation limit. Senescence had been considered as an anti-tumor strategy against cancer since its discovery because it stops cell proliferation. However, recent studies reveal that senescence does not necessarily prevent cancer but sometimes contributes to tumorigenesis.

Senescent cells have unique characteristics that result in a high activation of the cGAS-STING pathway, which detects cytosolic double-stranded DNA (dsDNA). The activated cGAS-STING pathway induces the transcription of the senescence-associated secretory phenotype (SASP), including interferons, which are eventually secreted from the cells [22,23,24,25,26,27,28,29]. Chronic activation of the cGAS-STING pathway results in a pro-tumorigenic cellular environment because SASP contains inflammatory components. As telomere shortening induces senescence, dysfunctional telomeres trigger the cGAS-STING pathway [30]. Interestingly, the activation of the cGAS-STING pathway by dysfunctional telomeres is mediated by mitochondrial distress, suggesting potential connections between telomeres and mitochondria [31]. Apart from the cGAS-STING context, cGAS plays a role in senescence. As the cGAS-STING pathway detects cytosolic dsDNA, cGAS and STING localize to the cytoplasm; however, during mitosis in proliferating cells, cGAS predominantly localizes to the nucleus and associates with chromatin DNA [27]. In the same study, cGAS deletion inhibited senescence more than STING deletion. Another study found that cGAS interacts with telomeres and protects them from shortening independently of the cGAS-STING pathway [32]. Therefore, cGAS may regulate senescence through mechanisms yet to be elucidated.

Cancer cells can utilize the cGAS-STING pathway by their telomere maintenance machinery. Telomerase reverse transcriptase (TERT) is a telomerase complex protein subunit. Although most somatic cells do not express TERT in humans, TERT mRNA is shuttled from cancer cells via exosomes and turns nonmalignant telomerase negative cells into telomerase positive cells [33]. The spontaneous immortalization of nonmalignant cells induced by TERT expression triggers the cGAS-STING pathway [27], suggesting that telomerase-positive cancer cells change their microenvironment to being pro-tumorigenic by transforming nonmalignant cells into being telomerase positive and activating the cGAS-STING pathway. Cancer cells with the other telomere maintenance mechanism, ALT, show a unique feature of extrachromosomal telomere repeats (ECTR) in the cytoplasm. This is derived from telomere-trimming activity and HR-mediated telomeres [34] and serves as an ALT template [35]. ECTR in normal cells activates the cGAS-STING pathway and facilitates the immune response, leading to a perturbation of proliferation, whereas ALT cells have defects in the cGAS-STING pathway and evade the antiproliferation effects [36]. The cGAS-STING deficiency depends on ATRX, Daxx, and H3.3, whose mutations have been found in cell lines associated with ALT [37,38]. Similar to the secretion of TERT mRNA, ECTR may activate the cGAS-STING pathway in nonmalignant cells by being shuttled from ALT cells, resulting in a pro-tumorigenic microenvironment. Even though further research is needed, previous studies have proposed the association between the telomere maintenance mechanisms and the cGAS-STING pathway, which contributes to cancer development. For a better understanding of machinery, we describe detailed mechanisms of each telomere maintenance and the cGAS-STING pathway in cancer regulation in the next chapters.

## 3. Cancer and Telomerase

Telomerase is a ribonucleoprotein complex that elongates telomeres to prevent telomere loss [39]. It is recruited to shortened telomeres during the S-phase and lengthens them by reverse transcription [40,41,42]. The primary components of human telomerase are a protein subunit TERT, an RNA subunit known as telomere RNA component (TR or TERC), and accessory proteins including dyskerin (DKC1) [43,44,45], GAR1 [45,46,47], NHP2 [45,48], NOP10 [45,48], and TCAB1 [49,50]. Telomerase activity during telomere elongation relies on the reverse transcription activity of TERT with the telomere-complementary RNA template, TERC. TERT and TERC are sufficient to reconstitute telomerase activity in vitro [51]; however, accessory proteins in the telomerase complex improve the efficiency of telomere maintenance. TERC has a 3′ H/ACA small nucleolar RNA-like domain, which contains the H box in the single-stranded hinge region after the 5′ hairpin and the ACA box at the 3′-terminal tail (Figure 2A). The H/ACA accessory proteins in the telomerase complex such as DKC1, GAR1, NHP2, and NOP10 interact with the H/ACA motif of TERC to stabilize it [52] and enhance the TERC–TERT interaction [53] (Figure 2A). The stabilization of TERC by these H/ACA proteins is achieved by the pseudouridylation of TERC [54]. Nevertheless, the molecular mechanisms for stabilization remain unclear because the H/ACA proteins interact with TERC at a position that is different from the well-studied pseudouridylation of ribosomal RNA by the H/ACA proteins [55,56]. Another telomerase accessory protein, TCAB1, enhances telomere maintenance in vivo by the proper recruitment of telomerase to the telomeres. TCAB1 increases the localization of the telomerase complex at the Cajal body, a membraneless organelle in the nucleus, which contains telomeres during the S-phase, by binding to the CAB-box motif of TERC as a Cajal body chaperone [49,50,57,58]. Consistent with studies showing the importance of TCAB1 in telomere regulation, the mutation or deletion of TCAB1 induces telomere shortening [50,59,60]. Recently, protein interactions of the telomerase complex with other DNA-binding proteins have been revealed. TPP1, one of the shelterin components, serves as a structured interface between the telomerase-essential N-terminal domain and the telomerase RAP motif of TERT [61]. In addition, from structural studies, a histone H2A–H2B dimer was identified as a holoenzyme that binds TERC [61,62]. The role of the histone heterodimer remains elusive; however, the findings suggest a noncanonical role for histones beyond the nucleosome that stabilizes and/or recruits telomerase to the chromosome ends.

In humans, TERT is specifically expressed in most immortalized cells, including 85–90% of all human cancers [63,64], germ cells, and stem cells. TERC is expressed broadly in most cells [63]. Thus, TERT expression is a hallmark of cancer cells that exhibit telomerase activity. However, TERT shows noncanonical functions other than telomere-related functions. The nuclear localization of TERT requires its binding to NF-κB p65 [65]. This binding of TERT with NF-κB p65 is considered to play a noncanonical role beyond telomere regulation, because the binding was also observed for a TERT mutant, TERT K626A, which lacks the activity of telomere reverse transcription [66]. The TERT–NF-κB p65 interaction increased a subset of NF-κB-dependent gene expressions such as IL-6 and TNF-α, which are critical for inflammation and cancer progression [65], suggesting a noncanonical role for TERT in cancer regulation. Another study of nuclear TERT showed that TERT activates vascular epithelial growth factor (VEGF) gene expression through the interaction with the VEGF gene promoter and the binding to the transcription factor Sp1 [67]. Inhibiting TERT expression reduced VEGF expression and tumor growth. From the studies, we can conclude that TERT plays a noncanonical role that regulates cancer development through different mechanisms from telomere regulation in the nucleus.

TERT localizes not only in the nucleus but also in the mitochondria [68,69,70,71]. Mitochondrial DNA is circular, lacks telomeres, and does not contain TERC [72], suggesting a noncanonical role for TERT in the mitochondria. Several reports indicate that mitochondrial TERT reduces mitochondrial DNA (mtDNA) damage [70,72,73], attenuates ROS generation [70,71,72,73,74,75], and enhances electron transport chain (ETC) activity [70,71,72,73,74]. Although the underlying mechanisms for these processes remain elusive, TERT increases antioxidant manganese superoxide dismutase (MnSOD) levels, which may contribute to the mitochondrial function of TERT [75,76]. The effects of mitochondrial TERT on mitochondria function are consistent with the finding that mitochondrial TERT inhibits mitochondrial apoptosis [70,71]. However, apoptosis regulation by mitochondrial TERT remains controversial as several studies also contend that mitochondrial TERT induces apoptosis [10,69]. To address this controversy, we simultaneously evaluated TERT localization and apoptosis, and discovered that mitochondrial TERT exhibits two opposing effects on apoptosis at different stages, which may explain the controversy [77]. TERT binds to nucleotides, such as mitochondrial DNA encoding ND1 and ND2 [73], as well as mitochondrial tRNA [72]. The interaction between TERT and nucleotides in the mitochondria may promote mitochondrial function. Most mitochondrial TERT studies have been conducted on telomerase-positive cancer cells. Thus, the regulation of mitochondrial TERT to enhance mitochondrial biogenesis may contribute to the aberrant proliferation and survival of cancer cells. Although most TERT studies focus on the full-length construct, TERT actually has at least 14 alternative splice variants [78,79,80,81,82]. A recent study found that TERT gene expression, including the alternative splice variants, is regulated by the length of the telomeres at chromosome 5p and this regulation is mediated by the sheltering component, TRF2 [83]. As most variants lack the telomerase activity domain, it is reasonable to assume that the TERT splice variants have a non-telomeric function. Some of these variants are highly expressed in cancer cells; one was even localized to the mitochondria and nucleus, and inhibited apoptosis [84]. Overall, TERT contributes to DDR, not only in the nucleus but also in the mitochondria (Figure 2B).

## 4. Cancer and ALT

Although most cancer cells exhibit the telomerase-dependent regulation of telomeres, the other 10–15% of human cancer cells [85] utilize a different telomere repair machinery, known as ALT. ALT was first discovered in immortalized human cell lines that contained long telomeres but lacked telomerase expression [9]. The recombination-based ALT mechanism was proposed in a study of telomerase-negative yeast cells [86]. The hypothesis of the ALT mechanism was confirmed when a DNA tag inserted into telomeres was copied to other telomeres in telomerase-negative human cells [87]. ALT assays using tumor cells revealed that ALT is likely to occur in tumors derived from mesenchymal origin, such as glioblastoma multiforme [88], osteosarcomas [89], and soft-tissue sarcomas [89,90,91], although the underlying mechanism remains elusive.

Proteins involved in ALT include those that participate in conventional DDR, especially during HR. When telomeres are exposed to replication stress and contain DSBs, the DDR cascade is activated and recruits initiators of end-resection at DSBs, such as the MRE-11, Rad-50, and NBS-1 (MRN) complex [92,93]. In HR, the MRN complex unwinds the DNA lesion and recruits ataxia telangiectasia mutated (ATM) kinase to the damaged telomeres through an interaction with NBS-1 [94,95]. Through binding to the MRN complex and unwinding DNA, ATM adopts a conformational change that induces autophosphorylation at Ser1981 [96]. The autophosphorylation of ATM changes it from an inactive ATM dimer to an active ATM monomer, which results in signal transduction via ATM kinase activity [96]. The active ATM monomer phosphorylates CtIP to activate the CtIP and MRN complex [97,98]. The CtIP–MRN complex creates 3′-overhangs on DNA by executing a 5′ to 3′-resection of the DNA ends [99,100]. These 3′-overhangs enable the strand invasion necessary for HR. As both ATM and CtIP are involved in telomere maintenance [101,102], it is tempting to speculate that the ALT pathway relies on the MRN complex-mediated HR machinery. In contrast, the active ATM monomer phosphorylates histone H2AX at the DNA lesion, which prevents CtIP-dependent end-resection [103,104]. Moreover, the active ATM monomer phosphorylates the DDR protein, p53-binding protein 1 (53BP1) [105], which inhibits HR and promotes nonhomologous end joining (NHEJ) [106]. Furthermore, γH2AX DNA damage foci and 53BP1 colocalize with telomere dysfunction-induced foci [107]; however, the role of 53BP1 in DDR remains controversial [108]. This suggests that future studies of γH2AX and 53BP1 focusing on ALT, rather than on HR or NHEJ, are warranted. During ALT, HR-associated Rad51 forms filaments between telomere foci and promotes the directional movement of the foci for interchromosomal homology searches [109]. This observation is consistent with the role of Rad51 in DDR other than telomeres [110,111,112]. Therefore, ALT relies on the DDR pathways, particularly the HR pathway, to repair telomeres.

When telomerase expression is inhibited in telomerase-positive cancer cells, the cells also engage in ALT [11]. These cells upregulate the expression of the proliferator-activated receptor-γ coactivator-1β (PGC-1β), a master regulator of mitochondrial biogenesis and function. In normal cells, telomere dysfunction suppresses PGC-1β and the PGC-1β-homolog, PGC-1α, by activating p53, which binds to their promoters [113]. The molecular mechanism underlying the mitochondrial improvement by ALT induction remains unknown, but it may be associated with p53 suppression, because most ALT cells lack a functional p53 [107]. In addition, the ALT–PGC-1β axis may rely on similar machinery for DDR-related cross talk between the nucleus and mitochondria. PGC-1α is also involved in the DDR machinery along with the DNA sensor, poly (ADP-ribose) polymerase 1 (PARP1). PARP1 recognizes the DNA break and initiates DNA repair by consuming NAD+. When PARP1 is inhibited, NAD+ is consumed by NAD-dependent deacetylase sirtuin-1 (SIRT1), which deacetylates several mitochondrial substrates, such as PGC-1α [114]. As PARP1 is associated with single-strand break repair, base-excision repair, and NHEJ, other than HR in DDR, PARP1 inhibition in HR-deficient cells, such as breast cancer gene 1/2 mutants, is effective in treating cancers [115,116]. In addition, PARP interacts with p53 to inhibit tumorigenesis [117]. Therefore, ALT cells may promote HR in telomeres and exhibit a similar PGC-1-activating effect from PARP1/p53 inhibition. This assumption is also supported by the finding that SIRT1 overexpression increases HR in the genome, including telomeres [118]. Overall, ALT appears to utilize the DDR pathways for cross talk between telomeres and the mitochondria (Figure 2B).

## 5. Cancer and the cGAS-STING Pathway

Besides the potential association between telomere maintenance machinery and the cGAS-STING pathway, cancer cells actually utilize the cGAS-STING pathway to promote cell-cycle progression. The cGAS-STING pathway is an evolutionarily conserved mechanism that detects intracellular DNA and mediates the inflammatory response to protect cells from infections. cGAS is a DNA-sensing enzyme that detects dsDNA and catalyzes the conversion of ATP and GTP into cyclic GMP–AMP (cGAMP) [119,120]. cGAMP binds to and activates an adaptor protein, STING, on the endoplasmic reticulum (ER) membrane [121,122]. Activated STING forms tetramers and translocates from the ER to the ER–Golgi intermediate compartment [123,124,125]. STING is palmitoylated at the Golgi apparatus and recruits TANK binding kinase 1 (TBK1) and interferon regulatory factor 3 (IRF3) [122,126,127]. TBK1 phosphorylates IRF3, which subsequently forms dimers, translocates to the nucleus, and drives the expression of immune and inflammatory mediators, including type-I and type-III interferons [128]. As cGAS is indiscriminate with respect to dsDNA sequences, this pathway is triggered not only by viral pathogens but also by self-DNA [120,129,130]. Unstable chromosomes mis-segregate during cell division, which results in micronuclei containing genomic contents. These micronuclei rupture and expose their genomic contents to the cytosol, which activates cGAS and the proinflammatory response. Along with this context, genome instability, a hallmark of cancer cells, generates cytosolic dsDNA that activates the cGAS-STING pathway and induces chronic inflammation [13,14]. Cancer cells also manage the cGAS-STING pathway when the micronuclei emerge from DNA damage. The micronuclei arise from exogenous DNA damage induced by radiation and activate the cGAS-STING pathway in nonmalignant cells [13,131]. Simultaneously, DNA damage also produces these micronuclei and triggers the cGAS-STING pathway in cancer cells [132]. However, when the radiation doses exceed 12–18 Gy, cancer cells induce the DNA exonuclease Trex1 and attenuate the cGAS-STING response by degrading DNA that accumulates in the cytosol upon radiation [132]. The reduction in the cGAS-STING response protects cancer cells from tumor rejection by CD8^+^ T cells. Thus, cancer cells benefit from the cGAS-STING pathway, which is activated by their innate genome instability and DNA damage induced by radiotherapy.

In addition to the genome, the mitochondria in cancer cells contribute to cGAS-STING activation. Mitochondrial nucleoids are higher-order structures involved in mitochondrial maintenance and gene expression [133]. Nucleoids anchor mtDNA through mitochondrial transcription factor A (TFAM) binding and cancer cells often contain TFAM mutations [134]. In mouse primary cell cultures, TFAM deficiency enables mtDNA to diffuse into the cytosol and activate the cGAS-STING pathway [135]. Moreover, cancer cells with mitochondrial dysfunction release mtDNA and the cells with STING exhibit impaired cellular fitness, suggesting an anti-tumorigenic effect of the cGAS-STING pathway in cancer [136]. In contrast, cancer cells may benefit from the pro-tumorigenic effects of the cGAS-STING pathway through mtDNA transfer from surrounding cells. Tumors lacking mtDNA can recover mtDNA from host cells [137]. Hormonal-therapy-resistant metastatic breast cancers acquire mtDNA from CAF-derived extracellular vesicles (EVs) [138]. As activated T cells release EVs containing mtDNA and activate the cGAS-STING pathway in dendritic cells [139], it is possible that cancer cells also activate the cGAS-STING pathway by importing mtDNA from EVs secreted by surrounding cells. Considering that the cell-autonomous release of mtDNA is associated with mitochondrial dysfunction, the pro-tumorigenic effects of the cGAS-STING pathway may depend on cytosolic mtDNA from extracellular sources without mitochondrial dysfunction. To prevent the anti-tumorigenic effects of the stressful combination of the cGAS-STING pathway and mitochondrial dysfunction, improving mitochondrial homeostasis with TERT and ALT may mitigate acute stress (Figure 3).

## 6. The cGAS-STING Pathway in Surrounding Cells

Similar to cancer regulation of the cGAS-STING pathway with EVs from surrounding cells, the cGAS-STING pathway inside the surrounding cells also contributes to tumor homeostasis. The tumor microenvironment (TME) enables precancerous cells to activate oncogenes, such as RAS [140,141]. The activation of oncogenes causes dysregulation of DNA replication and DNA damage, eventually leading to senescence, which is a state of irreversible cell-cycle arrest. This phenomenon was first defined as the proliferation limit induced by telomere shortening. In senescent cells, expression of the cyclin-dependent kinase (CDK) inhibitors p16INK4a and p21WAF1/CIP1 is upregulated. Inhibition of the cyclin D-CDK4/6 and cyclin E-CDK2 complexes by p16INK4a and p21WAF1/CIP1, respectively, suppresses E2F target genes and activates the cell-cycle checkpoint [142,143,144,145,146]. Senescent cells have unique characteristics that result in a high activation of the cGAS-STING pathway. They downregulate the E2F transcriptional targets, DNase2 and TREX1, which eliminate cytosolic dsDNA in normal cells [23]. They also downregulate lamin B, which is required for integrity of the nuclear envelope and prevents the release of chromatin fragments into the cytosol [22,147]. Finally, they upregulate long interspersed element-1, which is reverse-transcribed into cDNA in the cytosol [148,149,150]. In the TME, cancer and senescent precancerous cells transform surrounding noncancerous fibroblasts into CAFs, which secrete inflammatory factors similar to SASP, including small EVs, to promote cancer development and progression [151,152,153,154,155]. We previously demonstrated that senescent hepatic stellate cells (HSCs), which are induced by deoxycholic acid, a gut bacterial metabolite derived from altered gut microbiota in obesity, secrete IL-1β, one of the SASP factors, promoting the development of obesity-associated hepatocellular carcinoma (HCC) [151]. Moreover, senescent HSCs also secret prostaglandin E2 (PGE2), causing the suppression of anti-tumor immunity and the progression of obesity-associated HCC [152]. Besides the inflammatory protein secretion, our previous studies also established that small EVs’ secretion was significantly increased in senescent cells, and small EVs contain erythropoietin-producing hepatocellular receptor A2 (EpHA2) or pericentromeric non-coding RNA (Satellite II RNA) acting as a SASP factor, thereby provoking carcinogenesis [153,154,155]. Another study shows that cGAMP, the mediator of cGAS-STING pathway, is transferred from cells to cells through gap junctions, inducing SASP in cells that do not activate cGAS [156]. This cGAMP transfer would also contribute to spread the TME and evoke carcinogenesis.

In addition to oncogenes, treatment with chemotherapeutic drugs or ionizing radiation induces DNA damage or cell-cycle arrest in cells surrounding the cancer cells, resulting in cellular senescence [157,158]. This therapy-induced senescence impedes successful cancer intervention because it contributes to tumor recurrence by pro-tumorigenic changes in the TME (Figure 3). Specifically, chemotherapy induces senescence even in nonmalignant cells because of its lack of specificity, although chemotherapeutics target the high proliferative characteristics of tumors. Doxorubicin arrests DNA replication and induces senescence in cancer cells by activating DDR pathways resulting from topoisomerase inhibition [159,160], whereas it induces senescence with SASP secretion in nonmalignant cells [161,162,163]. Paclitaxel also induces senescence and SASP in nonmalignant cells, such as mouse dermal fibroblasts [163], whereas it arrests cells at mitosis by stabilizing polymerized microtubules and inhibiting the metaphase–anaphase transition [164]. Bleomycin, which damages DNA by oxidation [165], induces SASP secretion in skin fibroblasts and prostate stromal cells [166,167]. If only a fraction of the cancer cells survives these therapies, the SASP-mediated chronic inflammatory TME, affected by the surrounding cells, contributes to tumor recurrence.

To overcome the SASP effect induced by chemotherapy, senescent cell clearance (senolytics) and SASP inhibition (senomorphics) represent adjuvant strategies for cancer therapy. Senescent cells are resistant to apoptosis through the upregulation of antiapoptotic genes [168]. Therefore, blocking these antiapoptotic pathways eliminates senescent cells, but maintains proliferative cells. For example, ABT-263, also known as Navitoclax, is a mimetic of the proapoptotic protein BH3, and inhibits the antiapoptotic effect of the Bcl-xL and BAX interaction [169]. Administration of ABT-263 following doxorubicin treatment increases tumor suppression [170,171]. Furthermore, we recently reported that the BET family degrader, ARV825, is an effective senolytic drug that can specifically eliminate senescent cells. Treatment of obese mice with ARV825 inhibited hepatocellular carcinoma development by eliminating senescent hepatic stellate cells from the liver [172]. Another strategy involves SASP inhibition and is known as senomorphics. SASP secretion is mediated by signaling pathways, such as mTOR, p38MAPK, and NF-κB. Rapamycin, an inhibitor of the mTOR pathway, downregulates cancer cell proliferation by blocking SASP secretion by interfering with interleukin (IL) 1 alpha protein translation [173]. Furthermore, targeting the p38MAPK signaling protein, TAK1, using 5Z-7-oxozeaenol, decreases SASP and inhibits the pro-tumorigenic effect in patients after chemotherapy [174]. Anakinra, an antagonist of the IL-1 receptor, inhibits IL-6 and IL-8 secretion from senescent cells, thus decreasing the invasion of metastatic cancer cells [175]. Although senolytics and senomorphics may generate unexpected side effects because of the abundance of senescent cells and SASP in humans, the above studies demonstrate the potential benefit of combined anticancer therapy using strategies targeting senescence.

## 7. Future Perspectives/Conclusions

Over the past decade, studies have revealed that DDR systems in cancer have novel roles in contrast to how they were first discovered. TERT, a component of telomerase, exhibits a noncanonical function beyond telomere regulation in mitochondria. Mitochondrial TERT protects mtDNA, reduces ROS levels, enhances ETC function, and alters apoptosis. TERT interacts with mitochondrial nucleotides, such as mtDNA and mitochondrial tRNA, in the mitochondria. The interaction of TERT with these mitochondrial nucleotides, besides MnSOD, may regulate the noncanonical function of mitochondrial TERT. The other form of telomere maintenance, ALT, also involves mitochondrial function by increasing the activity of the mitochondrial regulator, PGC. In addition to telomere maintenance, ALT contributes to the cGAS-STING pathway. HR-mediated trimming of telomeres provides ECTR, which activates the cGAS-STING pathway. Although the cGAS-STING pathway was first described as a defense mechanism against pathogens, numerous studies indicate that the pathway also contributes to the pro-tumorigenic cancer environment by inflammatory SASP. The pro-tumorigenic effect of SASP arises not only from cancer cells themselves but also from surrounding stromal cells. This makes it difficult for cancer therapy that causes nonspecific damage to normal cells to completely cure cancer. Overall, cancer utilizes these DDR pathways to proliferate in situations in which normal cells do not function well.

Further studies are needed to target DDR-related pathways in cancer. The role of mitochondrial full-length TERT has been elucidated, but the molecular machinery for its mitochondrial role remains to be identified. With respect to the alternative splice variants of TERT, their relationship with cancer remains unclear. As most splice variants lack the TERT domain that interacts with nucleotides, it is likely that the splice variants have different regulatory mechanisms in cancer biology. The role of ALT in nuclear–mitochondria crosstalk also requires further study. The molecular understanding of these telomere-related DDR pathways in cancer biology will pave the way for future studies focused on the effect of these pathways on tumor progression. For the cGAS-STING pathway, in which ALT is also involved, a molecular understanding of the pathway is progressing; however, the manner in which acute or chronic activation of this pathway is regulated remains an open question. Besides, apparently promising senolytics, such as ABT-263, are not a universal solution to senescent cells in the TME. Considering the potential side effects of senolytics and senomorphics, future preclinical studies, including the use of ABT-263, will provide proof-of-principle for cancer treatment combined with a senescence-targeted strategy. A better understanding of these pathways will lead to future therapies and drugs to treat cancer and cancer-related diseases.

## Figures and Tables

**Figure 1 cells-11-01958-f001:**
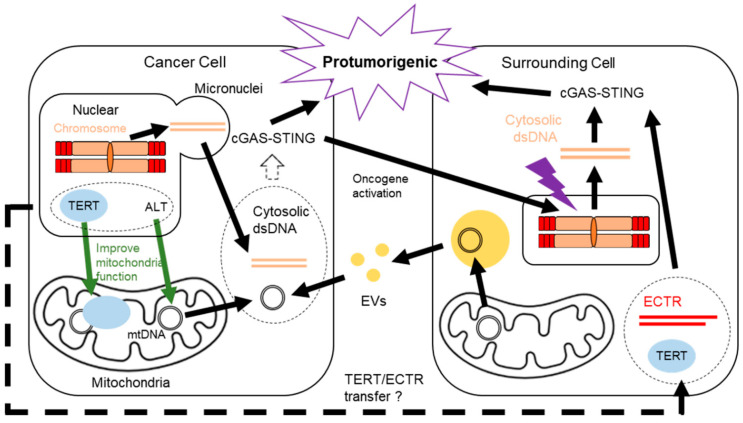
Overview of the connections between telomere maintenance and the cGAS-STING pathway in the nucleus and mitochondria of cancer cells. Cancer cells overcome the proliferation limit from telomere shortening by telomere maintenance mechanisms. Molecules involved in telomere maintenance also have noncanonical functions in the mitochondria. The mitochondrial functions of telomere-related molecules protect cancer cells from various stresses. The long-lasting cancer cells change the surrounding environment pro-tumorigenic by the activation of the cGAS-STING pathway in normal cells surrounding cancer cells, as well as in cancer cells.

**Figure 2 cells-11-01958-f002:**
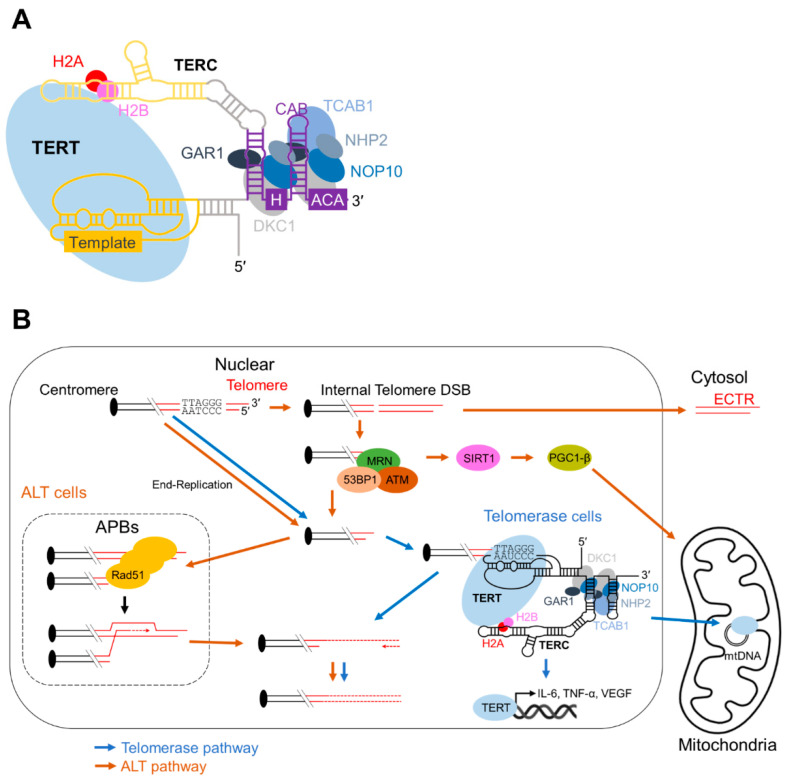
Telomere maintenance in cancer cells. Schematic images of (**A**) the telomerase complex and (**B**) pathways associated with telomerase and ALT in cancer cells. Cancer cells overcome telomere shortening by telomere maintenance mechanisms such as telomerase and ALT. In the telomerase complex, the telomerase RNA component TERC has the H/ACA motif and recruits the H/ACA proteins to stabilize the complex. In ALT, telomere lengthening is mediated by the HR-based DNA repair machinery. Molecules in the telomere maintenance also show the mitochondrial functions. In addition, cytosolic dsDNA fractions known as ECTR, which are produced from ALT, trigger the cGAS-STING pathway.

**Figure 3 cells-11-01958-f003:**
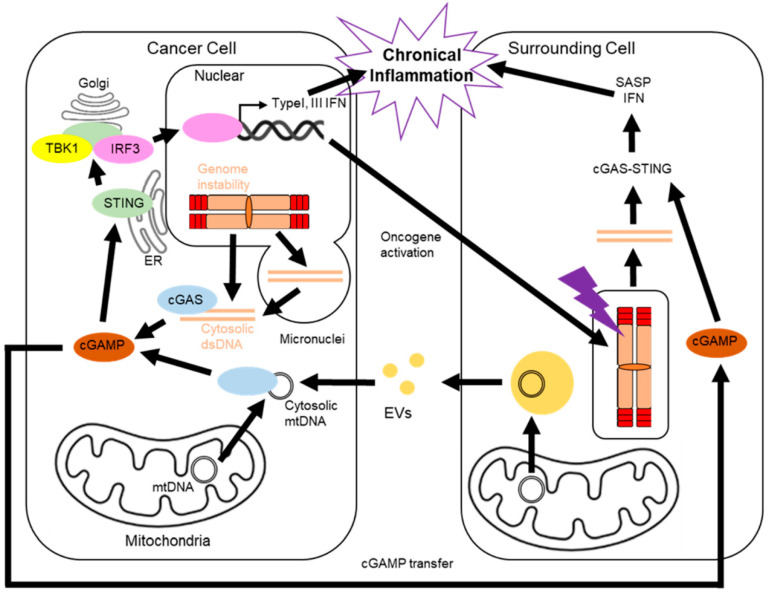
Overview of the cGAS-STING pathway in cancer cells and surrounding cells. Pro-tumorigenic TME in cancer from chronical inflammation by the cGAS-STING pathway in cancer cells and normal cells surrounding them. Genome instability and damaged mitochondria in cancer cells produce cytosolic dsDNA, which activates the cGAS DNA sensor. cGAS turns ATP and GTP to cGAMP, which activates STING on the ER membrane. The activated STING translocates to Golgi and recruits TBK1, which phosphorylates IRF3. The phosphorylated IRF3 promotes the expression of inflammatory factors, which results in the inflammatory environment. The inflammatory factors also activate oncogenes in normal cells surrounding the cancer cells, which triggers the cGAS-STING pathway in the normal cells. The chain reaction of inflammation creates the pro-tumorigenic TME in cancer.

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
