# Peer review of "Telomere Maintenance and the cGAS-STING Pathway in Cancer"

_cells, 2022, doi:10.3390/cells11121958_

Round 1

Reviewer 1 Report

The authors summarize the maintenance mechanisms of telomere and how cGAS/STING pathway activation is involved in tumorigenesis. Overall it is a thorough review with a lot of details yet descriptions are clear. Due to the importance of the topic, it will be of help for researchers in telomere, DNA damage and innate immunity. However, there are some concerns that would need to be addressed.

1.    It seems there is a disconnection between telomere and cGAS/STING regulation. Prior to reading the manuscript, I thought the authors would illustrate how telomere regulatory components may contributes to cGAS/STING activity control, or how dysregulation of telomere maintenance will lead to cGAS/STING activation. Current format only presents these two topics in separate sections- more connections and transitions would help to strengthen the manuscript.

2.    The review mostly focuses on how bystander cells regulates cGAS/STING signaling in cancer cells. Effects of DNA damage in cancer cells in addition to surrounding cells in regulating cGAS/STING signaling should also be included (eg line 319-332).

3.    cGAMP transfer should be included. eg. PMID: 24077100

4.    cGAS also plays a role in cellular senescence that would need to be discussed as well (PMID: 28533362)

Author Response

Point-by-point responses to the reviewers’ comments

We would like to thank all the three reviewers for their valuable comments and constructive suggestions. We have tried to address all the issues that they have noted and believe that the current manuscript has been significantly improved.

Reviewer #1

The authors summarize the maintenance mechanisms of telomere and how cGAS/STING pathway activation is involved in tumorigenesis. Overall it is a thorough review with a lot of details yet descriptions are clear. Due to the importance of the topic, it will be of help for researchers in telomere, DNA damage and innate immunity. However, there are some concerns that would need to be addressed.

  1. It seems there is a disconnection between telomere and cGAS/STING regulation. Prior to reading the manuscript, I thought the authors would illustrate how telomere regulatory components may contributes to cGAS/STING activity control, or how dysregulation of telomere maintenance will lead to cGAS/STING activation. Current format only presents these two topics in separate sections- more connections and transitions would help to strengthen the manuscript.

We agree with the reviewer and have added the chapter to describe the connecting telomere and cGAS/STING regulation (lines 73–128).

  1. The review mostly focuses on how bystander cells regulates cGAS/STING signaling in cancer cells. Effects of DNA damage in cancer cells in addition to surrounding cells in regulating cGAS/STING signaling should also be included (eg line 319-332).

Thank you for your comment. In line with your suggestion, we have added the sentences describing cGAS/STING and DNA damage in cancer cells (lines 288–298). In addition, we also cited relevant references (References No. 131, 132)

  1. cGAMP transfer should be included. eg. PMID: 24077100

Thank you for your suggestion. We have added the sentences describing cGAMP transfer and the reference (lines 354–357: References No. 156).

  1. cGAS also plays a role in cellular senescence that would need to be discussed as well (PMID: 28533362)

Thank you for your comment. We have described in more detail the role of cGAS plays in cellular senescence. (lines 99–105: References No. 27)

Reviewer 2 Report

This comprehensive review summarized the current understanding of the relationship between telomere/telomerase and cGAS-STING pathway in cancer. This review has two suggestions:

1.    Regarding the noncanonical function of telomerase, besides the role of hTERT in mitochondrial, hTERT plays significant roles in cancer development by interacting with transcription factor such as NF-kB p65, Sp1, I suggest to include this information.

2.    Telomere shortening or lengthening interacts directly with cGAS-STING pathway, I suggest to add a paragraph describe the relationship between telomere and cGAS-STING pathway.

Minor:  in the introduction section: “……By combining both mechanisms with an aberrant DNA damage response (DDR)….”. This sentence is not clear: I am not sure here “both mechanism” means “telomerase and ALT”? but telomerase and ALT usually mutually exclusive in cancer cells. 

Author Response

Point-by-point responses to the reviewers’ comments

We would like to thank all the three reviewers for their valuable comments and constructive suggestions. We have tried to address all the issues that they have noted and believe that the current manuscript has been significantly improved.

Reviewer #2

This comprehensive review summarized the current understanding of the relationship between telomere/telomerase and cGAS-STING pathway in cancer. This review has two suggestions:

  1. Regarding the noncanonical function of telomerase, besides the role of hTERT in mitochondrial, hTERT plays significant roles in cancer development by interacting with transcription factor such as NF-kB p65, Sp1, I suggest to include this information.

Thank you for your comment. We agree that this is an important point. In accordance with your suggestion, we have added the sentences describing the noncanonical function of nuclear TERT in the manuscript and the related references (lines 173–185: References No. 65, 66, 67).

  1. Telomere shortening or lengthening interacts directly with cGAS-STING pathway, I suggest to add a paragraph describe the relationship between telomere and cGAS-STING pathway.

We agree with the reviewer and have added the chapter to describe the connecting telomere and cGAS/STING regulation (lines 73–128).

Minor:  in the introduction section: “……By combining both mechanisms with an aberrant DNA damage response (DDR)….”. This sentence is not clear: I am not sure here “both mechanism” means “telomerase and ALT”? but telomerase and ALT usually mutually exclusive in cancer cells.

Thank you for your comment. To make the context clearer, we changed the word “both mechanisms” to “each telomere maintenance mechanism” (Line 41).

Reviewer 3 Report

Ebata and co-authors review the connection of telomere control with the cGAS-STING pathway. This is a well written review, the grammar and style is very good. I was not aware of the details on this connection but several key findings have been made over the past five years regarding these functions.

Overall the authors provide quite a bit of background information for each subtopic that is introduced. Many references and some rather unnecessary text book information is given when content related to DNA damage sensing, SASP, telomerase complex, tumor microenviroment are discussed. This is not wrong, but it dilutes the main and interesting connection discussed in the paper. For example the ECTR is first mentioned lined 232, and I think this is very central and should be more discussed and earlier. For example it can be seen already in figure 1 and figure 2. But the reader that is not in the field have no idea what it is. So I feel that there is too much information in parts of the paper and that masks, central interesting concepts. Maybe the authors can try to remedy this and center, focus the review further. And also not to be afraid of introducing key concepts and pathways rather early, also for the reader that is not in the field. Integrate the various sections more.

Minor comments:

Figure 1 and figure 3 takes up a lot of space in the manuscript, with very large fonts. I dont think that is necessary, they can be made smaller with less pronounced font, so that they match figure 2, that is more perfect and in style.

I'm not sure, but on line 115, I think it is meant to read telomerase and not telomeres (that localizes to Cajal bodies). Can the authors please check, it is important statement.

Author Response

Point-by-point responses to the reviewers’ comments

We would like to thank all the three reviewers for their valuable comments and constructive suggestions. We have tried to address all the issues that they have noted and believe that the current manuscript has been significantly improved.

Reviewer #3

Ebata and co-authors review the connection of telomere control with the cGAS-STING pathway. This is a well written review, the grammar and style is very good. I was not aware of the details on this connection but several key findings have been made over the past five years regarding these functions.

Overall the authors provide quite a bit of background information for each subtopic that is introduced. Many references and some rather unnecessary text book information is given when content related to DNA damage sensing, SASP, telomerase complex, tumor microenviroment are discussed. This is not wrong, but it dilutes the main and interesting connection discussed in the paper. For example the ECTR is first mentioned lined 232, and I think this is very central and should be more discussed and earlier. For example it can be seen already in figure 1 and figure 2. But the reader that is not in the field have no idea what it is. So I feel that there is too much information in parts of the paper and that masks, central interesting concepts. Maybe the authors can try to remedy this and center, focus the review further. And also not to be afraid of introducing key concepts and pathways rather early, also for the reader that is not in the field. Integrate the various sections more.

Thank you for your suggestions. We have agreed with the reviewer and have moved central concepts such as ECTR earlier in the manuscript (Line 115). We have also deleted some background information that could potentially dilute the central concepts such as shelterin components.

Minor comments:

Figure 1 and figure 3 takes up a lot of space in the manuscript, with very large fonts. I dont think that is necessary, they can be made smaller with less pronounced font, so that they match figure 2, that is more perfect and in style.

We agree with the reviewer and have made figure 1 and 3 smaller. In addition, we have added some changes to all the figures except for 2A to match the manuscript.

I'm not sure, but on line 115, I think it is meant to read telomerase and not telomeres (that localizes to Cajal bodies). Can the authors please check, it is important statement.

Thank you for your comment. We meant telomeres in the sentence that the reviewer pointed out, but we agree that the sentence was not clear and have changed the sentence for the sake of clarification (lines 149–152).

Round 2

Reviewer 2 Report

The  authors have addressed all my suggestions/comments to my satisfaction. The revised version has been much improved and I recommend the article for publication.